# A Rapid and Simple Assay Correlates In Vitro NetB Activity with *Clostridium perfringens* Pathogenicity in Chickens

**DOI:** 10.3390/microorganisms9081708

**Published:** 2021-08-11

**Authors:** Martina Hustá, Richard Ducatelle, Filip Van Immerseel, Evy Goossens

**Affiliations:** Livestock Gut Health Team (LiGHT) Ghent, Department of Pathology, Bacteriology and Avian Diseases, Faculty of Veterinary Medicine, Ghent University, Salisburylaan 133, B-9820 Merelbeke, Belgium; Martina.Husta@UGent.be (M.H.); Richard.Ducatelle@UGent.be (R.D.); Filip.Van.Immerseel@UGent.be (F.V.I.)

**Keywords:** in vitro activity assay, haemolysis, avian red blood cells, NetB, *Clostridium* *perfringens*

## Abstract

Necrotic enteritis is an important enteric disease in poultry, caused by NetB-producing *Clostridium (C.) perfringens* strains. As no straight-forward method to assess the NetB activity of *C. perfringens* was available, we aimed to develop an easy, high-throughput method to measure the NetB activity produced by *C. perfringens*. First, the appearance of *C. perfringens* on different avian blood agar plates was assessed. Based on the size of the haemolysis surrounding the *C. perfringens* colonies, NetB-positive strains could phenotypically be discriminated from NetB-negative strains on both chicken and duck blood agar. Additionally, strains producing the consensus NetB protein induced more pronounced haemolysis on chicken blood agar as compared to the weak outer haemolysis induced by A168T NetB-variant-producing *C. perfringens* strains. Next, a 96-well plate-based haemolysis assay to screen NetB activity in the *C. perfringens* culture supernatants was developed. Using this assay, a positive correlation between the in vitro NetB activity and virulence of the *C. perfringens* strains was shown. The developed activity assay allows us to screen novel *C. perfringens* isolates for their in vitro NetB activity, which could give valuable information on their disease-inducing potential, or identify molecules and (bacterial) metabolites that affect NetB expression and activity.

## 1. Introduction

The Gram-positive, spore-forming bacterium *Clostridium (C.) perfringens* is an anaerobic pathogen, known to cause a variety of histotoxic and enterotoxic diseases in humans and animals. The pathogen can produce more than 20 toxins [1,2]. Not all *C. perfringens* toxins are produced by all strains. Most of the toxin structural genes are encoded on large plasmids, one of which also includes the necrotic enteritis toxin B (NetB) [2,3]. NetB is the causative toxin for avian necrotic enteritis (NE), a disease that causes significant economic losses in the poultry industry worldwide due to compromised bird performance and increased morbidity and mortality [4,5,6]. To date, no effective non-antibiotic control strategies are available, although different control methods that could potentially prevent NE are being developed and are under investigation. These either intervene in the pathogenesis (e.g., control of predisposing factors, vaccines) or improve the microbiota–host interaction (e.g., dietary additives) [7,8]. While it is well established that the expression of virulence genes of a number of pathogens is affected by environmental factors and molecules, this area of research is under-explored for *C. perfringens* toxins [9,10,11,12].

To investigate the efficacy of a putative compound that might help controlling NE, time-consuming and expensive animal trials are usually required. These trials are necessary once a possible treatment is discovered. However, if the goal is to test whether a single compound might have a direct effect on NetB activity, a simple, rapid assay could provide a first indication on the efficacy of said compound. Traditionally, detection of NetB toxin activity is performed by an in vitro LMH cell-based toxicity assay [13,14]. This method is labour intensive and time consuming, and the induced cytotoxicity is a result of a combination of cytotoxic compounds secreted in the culture supernatants rather than NetB alone. More recently, Lee et al. described the development of an ELISA to detect the NetB toxin in both culture supernatants and biological samples [15,16]. Although such an ELISA is specific for NetB, no information on the biological activity is obtained. Therefore, there is a clear need for a simple, rapid test that specifically determines the NetB activity. Such an assay would allow us to screen novel *C. perfringens* isolates for their NetB activity, which could give valuable information on their disease-inducing potential. Additionally, it could be used to identify molecules and (bacterial) metabolites that affect NetB expression and activity. 

Haemolysis assays are commonly used for the detection and quantification of *C. perfringens* alpha toxin (CPA) and perfringolysin O (PFO) production in vitro. The specificity of these assays is linked to differences in the susceptibility of RBCs from various host species towards the respective toxins, with CPA activity being detected using sheep RBCs, whereas PFO is routinely detected using horse RBCs [17,18,19,20,21]. In the current study, we aimed to develop a simple, rapid, and cost-effective activity assay based on the haemolytic properties of NetB, that can be used in a high-throughput format to specifically measure NetB toxin activity in *C. perfringens* culture supernatants. The NetB activity assay was developed, based on the observation that NetB has haemolytic activity towards red blood cells (RBC) of different avian species, such as chicken, duck and goose [22]. The useability of the developed assay is shown by screening the NetB production of different *C. perfringens* isolates and comparing the NetB titre with the in vivo virulence of the strains.

## 2. Materials and Methods

### 2.1. Bacterial Strains and Growth Conditions

A collection of 9 NetB-positive and 5 NetB-negative *C. perfringens* strains isolated from broiler chickens was used (Table 1). Additionally, an alpha toxin mutant and a NetB toxin mutant from strain NE 18 were used (Table 1). All *C. perfringens* strains were grown on Columbia agar (Oxoid, Basingstoke, UK) in an anaerobic chamber with atmosphere containing 10% H_2_, 10% CO_2_ and 80% N_2_ (Jacomex, Dagneux, France). The identity of the NetB protein (consensus NetB or A168T variant NetB, [23,24]) was determined by amplifying a 448 bp region of the *netB* sequence spanning the region coding for the AA substitution (nucleotide substitution at position 502) using primers netB(186)FW: 5′-TGATACCGCTTCACATAAAGGT-3′ and netB(612)REV: 5′-ACCGTCCTTAGTCTCAACAAAT-3′, and Sanger sequencing of the obtained PCR amplicons by Eurofins Genomics (Ebersberg, Germany). The numbers included in the primer name indicate the 5′ position of the sense strand relative to the start codon (TTG) of the *netB* gene. *C. perfringens* strains were cultured anaerobically for 24 h at 37 °C in TGY broth (3% tryptone (Sigma Aldrich, St. Louis, MO, USA), 2% yeast extract (Sigma Aldrich), 0.1% glucose (Sigma Aldrich) and 0.1% L-cysteine (Sigma Aldrich)) unless stated otherwise. To determine the effect of the culture medium on the *C. perfringens* NetB toxin production, *C. perfringens* TGY overnight (ON) cultures were diluted 1/1000 in fresh TGY broth, BHI broth (VWR, Belgium) or minimal medium (MM; 50% tryptic soy broth (Sigma Aldrich), 25% nutrient broth (Oxoid) and 25% peptone water (Oxoid)), followed by anaerobic incubation for 24 h at 37 °C. To monitor the bacterial growth and toxin production, samples were taken every hour for seven hours including a last time point after 24 h. Strains remained under anaerobic conditions during the whole time period. Bacterial growth was determined by plating serial dilutions of the cultures on Columbia agar plates, followed by ON incubation at 37 °C under anaerobic conditions (Appendix A). Cell-free supernatants from the *C. perfringens* cultures were obtained by centrifugation at 4500× *g* for 5 min at 4 °C, followed by filtration of the supernatants through a 0.2 µm filter, and stored at −20 °C.

### 2.2. Recombinant NetB Production

Mature recombinant NetB, without its native signal peptide, was expressed in *Escherichia coli* using the pBAD TOPO^®^ TA Expression Kit (Invitrogen, Paisley, UK). A fragment encoding the *C. perfringens* NetB toxin (consensus *netB* gene; GenBank accession number EU143239.1) was amplified from the DNA of *C. perfringens* strain CP56 by PCR using a DNA polymerase with proofreading activity (Accuzyme, Bioline, Randolph, MA, USA). The forward primer (5′-GTTCTTGAGGAATAATAAATGAGTGAATTAAATGACATAAAC-3′) contained an in-frame stop codon and translation re-initiation sequence to remove the N-terminal leader and allow native protein expression. The reverse primer (5′-GTTCTTCAGATAATATTCTATTTTATGATCTTGCCAATT-3′) excluded the native *netB* gene stop codon. The resulting PCR product was incubated with Taq polymerase for 10 min at 72 °C (5 U; Promega, Madison, WI, USA) to add 3′ A-overhangs, cloned into the pBAD-TOPO expression vector and transformed into One Shot TOP10^®^ *E. coli*. The correct orientation of the *netB* toxin insert was verified by Sanger sequencing. Recombinant *E. coli* carrying the pBAD-NetB expression vector was grown at 37 °C and 125 rpm for 3 h until an OD_600_ of ±0.5 in Terrific Broth supplemented with 100 µg/mL ampicillin. Once the desired OD was reached, protein expression was induced with 0.002% L-arabinose (Sigma Aldrich) and the culture was further incubated ON at 37 °C and 125 rpm. Bacteria were harvested by centrifugation at 100,000 × *g*, 4 °C for 10 min (Beckman Coulter, Avanti J-E Ultracentrifuge), and pellets were subjected to one freeze–thaw cycle (−20 °C), followed by enzymatic lysis using BugBuster (Invitrogen). NetB protein was purified using a Ni-sepharose column (HisGravisTRap, GE Healthcare, Chicago, IL, USA) according to the manufacturer’s instructions. Finally, the purified protein was dialysed ON at 4 °C in phosphate-buffered saline containing 10% glycerol. Protein concentration was measured using BCA protein assay (Thermo Fisher Scientific, Waltham, MA, USA) and purity was assessed with SDS-PAGE (Figure 1). The produced rNetB was subsequently used to test its activity against chicken RBCs in a 96-well format.

### 2.3. Assessing the Potential of Avian Blood Agar Plates to Discriminate between NetB-Negative and NetB-Positive C. perfringens Strains

To determine whether NetB-positive and NetB-negative *C. perfringens* strains can be distinguished from each other based on the haemolytic activity towards blood from different avian species, an agar diffusion assay was performed. Therefore, Columbia blood agar (CBA) was prepared using Columbia blood agar base (Oxoid, Basingstoke, UK) supplemented with 5% of either goose, duck or chicken blood. Blood was added after sterilisation of the agar base, followed by dispersing the medium in 120 × 120 mm Petri dishes, air drying at room temperature and storing at 4 °C for maximum 30 days. The haemolytic profile after direct growth of *C. perfringens* on the CBA agar plates was determined by transferring the *C. perfringens* strain from an ON culture to the agar plates with a sterile toothpick. Additionally, the haemolytic activity in the supernatants of *C. perfringens* strains grown ON in TGY was determined. Therefore, small holes were pierced into the agar with the rear end of 20–200 µL pipette tips (diameter 7 mm) which were filled with 20 µL of supernatant per strain. Plates were incubated anaerobically overnight at 37 °C and subsequently scanned with a GS-800 calibrated densitometer (Bio-Rad Laboratories, Hercules, CA, USA). The diameter of the different haemolysis zones was measured using Quantity One software (Bio-Rad Laboratories).

### 2.4. Determination of the C. perfringens Haemolytic Activity towards Chicken Erythrocytes in a 96-Well Plate-Based Format

Chicken RBCs were obtained from whole blood by centrifugation at 1500× *g* for 10 min at room temperature (RT), after which the supernatant was removed and RBCs were gently resuspended in Hank’s Balanced Salt Solution (HBSS, Thermo Fisher Scientific). After another two washing steps with HBSS, the RBCs were diluted to a final concentration of 2% RBCs in HBSS (*v*/*v*). The haemolytic activity of the rNetB towards chicken RBCs was confirmed by incubating serial two-fold dilutions of the rNetB (starting from 40 µg/mL, diluted in HBSS) with an equal volume of 2% chicken RBCs. To determine whether chicken RBC haemolysis could be used as a measure to discriminate between NetB-positive and NetB-negative *C. perfringens* strains, equal volumes of chicken RBCs were incubated with 20% culture supernatants. In each assay, RBCs incubated with HBSS were used as a negative control (0% haemolysis), whereas RBCs diluted in distilled water (dH_2_O) were used as positive control (100% haemolysis). After incubation of the 96-well microtiter plates at 37 °C for 30 min, the plates were centrifuged to pellet intact red blood cells (1 min, 1000× *g*, RT). The supernatants were transferred to a new 96-well microtiter plate and the OD_550nm_ was determined (Multiskan GO, Thermo Scientific). Haemolytic activity was observed by the increase in absorbance due to the release of haemoglobin from the erythrocytes. 

### 2.5. Assessing the NetB-Specific Haemolytic Activity in C. perfringens Supernatants by Blocking Alpha Toxin and Perfringolysin-Induced Haemolysis

In addition to the NetB toxin, *C. perfringens* also produces other haemolytic toxins, namely alpha toxin (CPA) and perfringolysin O (PFO). In order to determine the NetB-specific activity in the *C. perfringens* culture supernatants, the haemolytic activity of CPA and PFO should be neutralised. Therefore, the activity of CPA and PFO was neutralised by incubating the culture supernatants with polyclonal antiserum from calves immunised with a native NetB-negative *C. perfringens* toxin preparation [29]. The CPA- and PFO-neutralising capacity of the antiserum was previously published [29]. In order to determine the amount of antiserum needed to completely block the CPA- and PFO-induced haemolysis towards chicken RBCs, a two-fold dilution series of the antiserum (20%–0.4% antiserum) was incubated with a constant amount of *C. perfringens* culture supernatants (20% SN), for 30 min at 37 °C prior to the addition of an equal volume of 2% chicken RBCs. *C. perfringens* culture supernatants without antiserum were used as a control. All measurements were performed in duplicate (technical replicates), using three independent biological replicates, consisting of supernatants from different ON cultures as well as blood from different birds.

### 2.6. Final Doubling Dilution Assay Protocol to Determine the NetB Titre in C. perfringens Culture Supernatants

In order to determine the NetB titre in *C. perfringens* culture supernatants, a doubling dilution assay protocol was established. Therefore, non-NetB haemolytic activities were blocked by incubating a two-fold dilution series of *C. perfringens* culture supernatants (ranging from 20 to 0.04% SN, diluted in HBSS) with an equal volume of antiserum directed towards both CPA and PFO (0.5% final concentration). This *C. perfringens* supernatants–antiserum mixture was incubated at 37 °C for 30 min, after which an equal volume of 2% pre-warmed chicken RBCs was added to each well, followed by 30 min incubation at 37 °C. Chicken RBCs diluted in dH_2_0 or HBSS were used as, respectively, a positive control (100% haemolysis), or negative control (0% haemolysis). For the calculation of the NetB titres for all strains, the blank (mean value of all wells containing negative control) was subtracted from all other OD values, after which a Hill curve was fitted to the concentration–response data to calculate the LD50 values for each strain, which represent the supernatant dilution that causes 50% haemolysis. The assay was performed in technical duplicate, for three biological replicates.

A detailed protocol to raise antisera towards *C. perfringens* CPA and PFO is added in Appendix A. The protocols to determine the concentration of antiserum needed to block non-NetB haemolytic toxins can be found in Appendix A, whereas the final protocol to determine the NetB titre is present in Appendix A. 

### 2.7. Detection of Alpha Toxin Activity

To determine the alpha toxin activity in the supernatants of strains JIR12058, EHE-NE18, CP56, JIR4866 and JIR4860, the lecithinase activity was assayed in an egg yolk agar well diffusion assay [30,31,32]. Therefore, small holes were pierced into Columbia agar supplemented with 2% egg yolk (*v*/*v*) with the rear end of a 20–200 µL pipette tip (diameter 7 mm), which were filled with 20 µL of supernatant per tested strain. A standard was included on each plate by preparing a two-fold dilution series of Phospholipase C (Sigma Aldrich) ranging from 1 to 0.0313 U/mL. Plates were incubated anaerobically overnight at 37 °C and scanned with a GS-800 calibrated densitometer (Bio-Rad Laboratories, Hercules, CA, USA). Alpha toxin activity was indicated by the development of turbidity. The area of the opaque zones was measured using ImageJ software (U.S. National Institutes of Health, Bethesda, MD, USA).

### 2.8. Statistical Analysis

Statistical analysis was performed using GraphPad Prism 8.4.3. For not normally distributed values, a Mann–Whitney U test (2 groups) or Kruskal–Wallis test (>2 groups) was performed to identify significant differences between groups, otherwise an independent *t*-test (2 groups) or ANOVA (>2 groups) was employed to identify significant differences between groups. For comparison of the NetB titres of different *C. perfringens* strains grown in different culture media, a two-way ANOVA was preformed, followed by a Tukey’s multiple comparison test. Pearson correlation was applied to assess whether there was a link between the NetB titres and the previously published virulence of the *C. perfringens* strains [23]. As the alpha toxin activity was not normally distributed, a Spearman correlation was applied to determine the correlation between the alpha toxin activity and either the NetB titre or virulence of the *C. perfringens* strains. For all analyses, *p*-values smaller than 0.05 were considered statistically significant.

## 3. Results

### 3.1. Assessing the C. perfringens Haemolysis Patterns on Avian Blood Agar Plates

Colonies of a NetB-positive *C. perfringens* strain (NE18) as well as its NetB mutant (NE18 Δ*netB*), and its alpha toxin mutant (NE18 Δ*α*) were grown on Columbia agar supplemented with either chicken, duck or goose blood (Figure 2 and Appendix A). The wild-type NE18 strain induced a double haemolysis zone on all avian blood types, while the NetB mutant (NE18 Δ*netB*) did not induce an outer haemolysis zone on any of the tested avian blood agars (Figure 2A,C,E), indicating that the outer haemolysis zone might be caused by NetB. However, no zone with complete clearing around the colonies was observed, with both the inner and outer haemolysis zone showing only partial haemolysis on all avian blood agar types. This makes it difficult to differentiate between an inner and outer haemolysis zone, an effect that was most pronounced on chicken and duck agar (Appendix A). When the alpha toxin mutant was grown on CBA supplemented with chicken blood, no boundary between the inner and outer haemolytic zone could be observed, and only the outer haemolysis was measured (Figure 2A). It was, however, possible to measure the inner haemolytic zone against duck and goose blood (Figure 2C,E), indicating that this inner haemolytic zone is likely induced by a combination of other haemolysins, such as perfringolysin O.

In addition to growing the *C. perfringens* strains directly on the avian blood agar plates, supernatants from *C. perfringens* ON cultures grown in TGY were tested on the plates. In contrast to the situation where the *C. perfringens* colonies were grown on the agar, the culture supernatants did induce a narrow zone of complete haemolysis on all avian blood agar plates (Appendix A). Additionally, on chicken blood agar, both supernatants from the wild-type NE18 strain, as well as from the alpha toxin mutant strain (NE18 ∆*α*), were able to induce a subtle, weak outer haemolysis zone. However, supernatants of the NetB mutant strain (NE18 ∆*netB*) did not induce an outer haemolytic zone on chicken blood agar (Figure 2B), indicating that the outer haemolytic zone is caused by NetB. On plates with duck and goose blood, no outer haemolysis was induced by any of the supernatants (Figure 2D,F), whereas the complete inner haemolytic zones were induced by all strains on all blood types, even by supernatant from the alpha toxin mutant (Figure 2B,D,F and Appendix A), an observation which further suggests activity of other haemolysins in the supernatant besides alpha toxin. 

### 3.2. NetB-Positive Strains Show Stronger Haemolysis on Avian Blood Agar Plates as Compared to NetB-Negative Strains

To further assess the possibility to discriminate NetB-positive from NetB-negative strains based on the haemolysis pattern on avian blood agar, a collection of six NetB-positive and four NetB negative strains was grown on Columbia agar plates supplemented with either chicken, duck or goose blood. Given the above-mentioned difficulty to differentiate the inner and outer haemolysis zone, only the largest haemolysis zone from each strain was measured (Figure 3). For both chicken and duck blood, a significant size difference of the lysis zones between the NetB-positive and NetB-negative strains was observed (chicken blood agar: *p* < 0.0001, Figure 3A,B; duck blood agar: *p* = 0.0006, Figure 3C,D). However, on chicken blood agar, a clear variation in haemolysis induced by the NetB-positive strains was observed. The *C. perfringens* strains CP56 and NE18 (both harbouring the consensus *netB* gene) induced a pronounced partial outer haemolysis zone, whereas for the NetB-positive strains JGS4100, D3, CP23 and S36 (all harbouring the G502A variant *netB* sequence), only a weak outer haemolysis zone was observed (Figure 3A). This variability in haemolytic pattern induced by the NetB-positive strains was similar, but less pronounced on duck or goose blood agar (Figure 3C,E). Additionally, on goose blood agar, a large variation in the size of the outer lysis zones caused by NetB-positive strains was observed, which could not be differentiated from the NetB-negative strains (*p =* 0.074, Figure 3E,F). 

### 3.3. Assessing the Haemolytic Activity of NetB in C. perfringens Supernatants and rNetB against Chicken RBCs

Based on the results from the avian blood agar experiments, chicken blood was selected to further develop a 96-well plate-based haemolysis assay. The haemolytic activity of NetB towards chicken erythrocytes was confirmed by incubating the chicken RBCs with a doubling dilution series of rNetB (Figure 4A). Haemolysis was observed as an increase in absorbance due to the release of haemoglobin from the RBCs. At the highest concentration (40 µg/mL), the rNetB was able to induce 88.39% ± 16.63% haemolysis of the chicken RBCs, which remained stable up to the concentration of 10 µg/mL rNetB (corresponding to 81.88% ± 7.71% haemolysis). When 5 µg/mL rNetB was incubated with the chicken RBCs, only 42.12% ± 3.58% haemolysis was observed, whereas from the dilution of 1.25 µg/mL rNetB downwards no haemolysis was induced (Figure 4A). Since the goal was to develop an assay where the NetB activity in *C. perfringens* supernatants could be assessed and compared, the haemolytic activity present in the culture supernatants of the NetB-positive *C. perfringens* strain (NE18), as well as its NetB mutant (NE18 Δ*netB*) and its alpha toxin mutant (NE18 Δ*α*), was assessed. No significant difference could be observed between NE18 WT and NE18 Δ*α* (*p* = 0.969), an indication that the haemolysis caused by the alpha toxin mutant is comparable with the haemolysis induced by the wild-type strain (Figure 4B). The NetB mutant induced significantly less haemolysis compared to the alpha toxin mutant (*p* = 0.004) and the NE18 wild-type strain (*p* = 0.007), suggesting that most haemolysis is indeed caused by NetB. Furthermore, when the alpha toxin and perfringolysin O activity of the NE18 NetB mutant strain was blocked by pre-incubating the culture supernatants with an excess of anti-CPA/PFO, no haemolysis could be observed (Figure 4B). 

### 3.4. Inhibition of Non-NetB Haemolysins Is Needed to Quantify NetB Activity in C. perfringens Culture Supernatants

To investigate the possibility to discriminate NetB-positive from NetB-negative *C. perfringens* strains based on their haemolytic activity towards chicken RBCs in a 96-well plate-based format, the haemolytic activity present in the culture supernatants of the strains was tested (Figure 5). Culture supernatants from the NetB-positive strains showed a significantly higher haemolytic activity towards chicken RBCs as compared to the NetB-negative strains (*p* = 0.0028). However, quite some variation in haemolysis induced by the NetB-negative strains was observed, making it hard to distinguish between a NetB-positive or NetB-negative isolate solely based on its haemolytic activity towards chicken RBCs. This clearly shows the need to block other haemolytic toxins present in the culture supernatants. 

The optimal concentration of the anti-CPA/PFO antiserum was determined by pre-incubating the *C. perfringens* culture supernatants with a dilution series of the antiserum prior to performing the haemolysis assay (data not shown). At an antiserum concentration of 0.1%, the haemolytic activity in the culture supernatants from all tested NetB-negative strains was completely blocked (Figure 5, grey), and the difference between the NetB positives and NetB negatives became even more significant (*p* < 0.0001). While pre-incubation of *C. perfringens* culture supernatants with anti-CPA/PFO antiserum significantly reduced the haemolysis caused by the NetB-negative strains (*p* = 0.0022), the antiserum had no effect on the haemolytic activity of the NetB-positive strains (*p* = 0.4601). Nevertheless, these results clearly show that alpha toxin and perfringolysin O have substantial haemolytic activity towards chicken erythrocytes, which might lead to false results when the goal is to only assess NetB activity. To ensure complete inhibition of CPA/PFO-induced haemolysis, a final serum concentration of 0.5% was therefore included in the final assay protocol (final protocol described in detail in Appendix A). 

### 3.5. Maximal NetB Toxin Activity Was Observed in the Late Stationary Phase

As the bacterial growth medium can have an effect on both the growth and toxin production of *C. perfringens*, the behaviour of four NetB-positive strains (CP56 and NE18 producing the consensus NetB protein; JGS4100 and D3 producing the A168T variant NetB protein) was assessed in three commonly used growth media (MM, TGY and BHI) (Appendix B). Maximal NetB toxin activity was observed in the late stationary phase (overnight culture). The growth medium did not have a significant effect on NetB activity. However, when using BHI medium, a high variability in NetB activity between the different replicates was observed, indicating that this culture medium is less suited to determine the NetB activity (Figure A1). TGY medium was the most sensitive to observe differences in NetB activity between the different *C. perfringens* strains. Therefore, further experiments were performed using late stationary cultures (overnight cultures) grown in TGY. 

### 3.6. The In Vitro NetB Production Correlates with C. perfringens Virulence

As NetB is essential in the pathogenesis of avian necrotic enteritis, the in vitro NetB production of five *C. perfringens* strains was compared to the virulence of these strains in an NE in vivo model. The virulence of each strain was previously reported as the average lesion score induced by each strain, compared to the average lesion score induced by strain NE18 in the same in vivo trial (NE18: reference strain included in all trials) [23]. The selected strains were grown in TGY medium for 24 h, after which the NetB titre in the culture supernatants was determined. To obtain a more complete overview of the toxin production by the strains, the in vitro alpha toxin activity was also measured (Table 2). 

The virulence of the *C. perfringens* strains was positively correlated with the in vitro NetB titre (Pearson *r* = 0.92; *p* = 0.028) but showed a trend towards a negative correlation with the in vitro alpha toxin production (Spearman *r* = −0.90; *p =* 0.083) (Figure 6). However, this inverse correlation between the virulence of the strains and in vitro alpha toxin production is mainly due to the high alpha toxin levels of a single, low-virulence strain (JIR4860). 

## 4. Discussion

Necrotic enteritis is an important enteric disease in poultry, causing significant economic losses to the livestock industry worldwide. *C. perfringens* NetB toxin is a major virulence factor in avian necrotic enteritis, and as a consequence, many studies directed towards understanding disease pathogenesis or the development of control strategies are focused on NetB. Identification of NetB-positive strains is routinely performed by amplification of the *netB* gene using PCR. However, there is no absolute correlation between the presence of the *netB* gene and the ability to produce NetB [15,24]. Furthermore, detection of the *netB* gene does not provide information about the NetB activity of the respective strains. In this study, we showed that NetB-positive strains can phenotypically be discriminated from NetB-negative strains based on the induced haemolysis of either chicken blood or duck blood agar plates. Indeed, on chicken or duck blood agar, NetB-positive strains show a large double haemolysis zone, which is in contrast to NetB-negative isolates, which induce a smaller, single haemolysis zone. However, on chicken blood agar, the NetB-positive strains could cause two different types of haemolysis. The *C. perfringens* strains NE18 and CP56 induced an easy to observe, pronounced partial outer haemolysis zone, whereas the four other tested strains induced a double haemolysis zone, where the outer haemolysis zone was remarkably weaker as compared to the outer haemolysis induced by NE18 or CP56. Remarkably, the strains inducing the stronger outer haemolysis (NE18 and CP56) are known to harbour the consensus *netB* gene, whereas the four strains showing weak outer haemolysis produce an alternative NetB toxin showing a single amino acid (A168T) difference. Previous research has shown that culture supernatants from different *C. perfringens* strains producing the A168T NetB variant were equally cytotoxic towards LMH cells as culture supernatants from strain NE18 [23]. However, this LMH cytotoxicity assay has a limited dynamic range and was shown to be less suited to screen the biological activity of NetB mutants [33]. Moreover, previous research from Savva et al. clearly showed that some synthetic AA substitutions (e.g., Y78A, Y187A, H188A) had no effect on the NetB-induced cytotoxicity towards LMH cells, while a significant reduction in haemolysis towards human RBCs was observed [33]. Although the naturally occurring A168T NetB variant is widespread (ranging from 26% (6/23) to 78% (21/27) of tested NetB-positive strains [23,24]), no research on the haemolytic potential of the A168T NetB variant is currently available. However, this AA variation is located in the pre-stem region of the NetB toxin, which is inserted in the lipid bilayer during pore formation, suggesting a possible role during pore formation and subsequent haemolysis [22,33]. Whether or not the NetB A168T variant is responsible for the weaker outer haemolysis zones induced by the NetB-positive strains in our study remains to be elucidated using recombinant protein. Furthermore, it is unclear why this difference in haemolytic profile observed for the strains grown on the chicken blood agar was not linked to a difference in haemolysis induced by the culture supernatants of the same strains towards chicken RBCs in solution. NetB production is positively regulated by the VirS/VirR two-component system [34]. However, it could be that other factors produced by *C. perfringens* might influence the expression of the *netB* gene or NetB activity itself, such as “VirR regulated RNA” (VR-RNA), an RNA molecule that has a regulatory function in Clostridia [34]. Therefore, the differences in NetB activity among the pathogenic strains might be the result of a combination of differences in gene expression and other influencing factors.

In this study, we present a rapid and easy 96-well plate-based assay to screen the NetB activity in *C. perfringens* culture supernatants. The assay is based on the high susceptibility of chicken RBCs towards the haemolytic activity of NetB toxin [22,33]. Although this haemolysis assay has previously been used to characterise the activity of various purified NetB toxin variants, our results clearly indicate that the haemolysis of chicken RBCs induced by *C. perfringens* culture supernatants is not specific for NetB toxin, and toxin-neutralising antisera are needed to block other haemolytic toxins produced by the *C. perfringens* strains. As TGY medium was the most sensitive to observe variations in NetB toxin production and the activity of different *C. perfringens* strains, we propose TGY as a suitable bacterial growth medium to assess NetB toxin production and activity. 

Various factors can affect the in vitro toxin production and, therefore, in vitro NetB production and activity cannot be directly linked to in vivo NetB production and subsequent pathogenicity. Therefore, in this study, we used the newly developed NetB assay to screen a selection of NetB-positive strains for their in vitro NetB production and linked this to the previously published pathogenicity of the strains in an NE in vivo model. The in vitro NetB activity was positively correlated with the virulence of the *C. perfringens* strains, further underscoring the importance of the NetB toxin in NE pathogenesis. This is in accordance with the previously reported observation that NetB-positive strains from NE flocks were able to produce NetB in vitro, whereas for the majority of the NetB-positive strains isolated from healthy broilers, no in vitro NetB production was observed using Western blot [24]. However, as only a limited number of strains with known virulence were available for this study, further research should focus on comparing a larger collection of *C. perfringens* strains to confirm these results. No link between the in vitro alpha toxin production and in vivo pathogenicity of the strains was observed, thereby confirming the results from previous studies comparing *C. perfringens* strains isolated from diseased and healthy poultry flocks [27,35,36].

## 5. Conclusions

The aim of this study was to develop a fast and easy method by which toxin production can be assessed by the measurement of haemolysis induced by NetB in supernatants. This protocol can be very helpful when it comes to further research about different compounds that might inhibit its activity, but also to evaluate which *C. perfringens* strains produce more active NetB. 

## Figures and Tables

**Figure 1 microorganisms-09-01708-f001:**
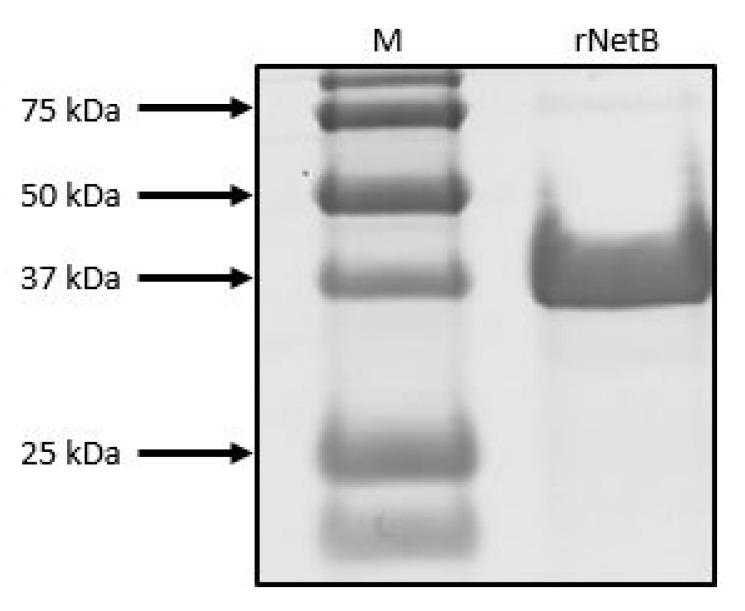
SDS-PAGE of purified recombinant NetB. The purified protein (rNetB) was separated by 12% SDS-PAGE and Precision Plus Protein™ Standard (BioRad) was used as a size marker (M). SDS-PAGE was stained with Brilliant Blue G; band representing rNetB appears at the expected size of 33 kDa.

**Figure 2 microorganisms-09-01708-f002:**
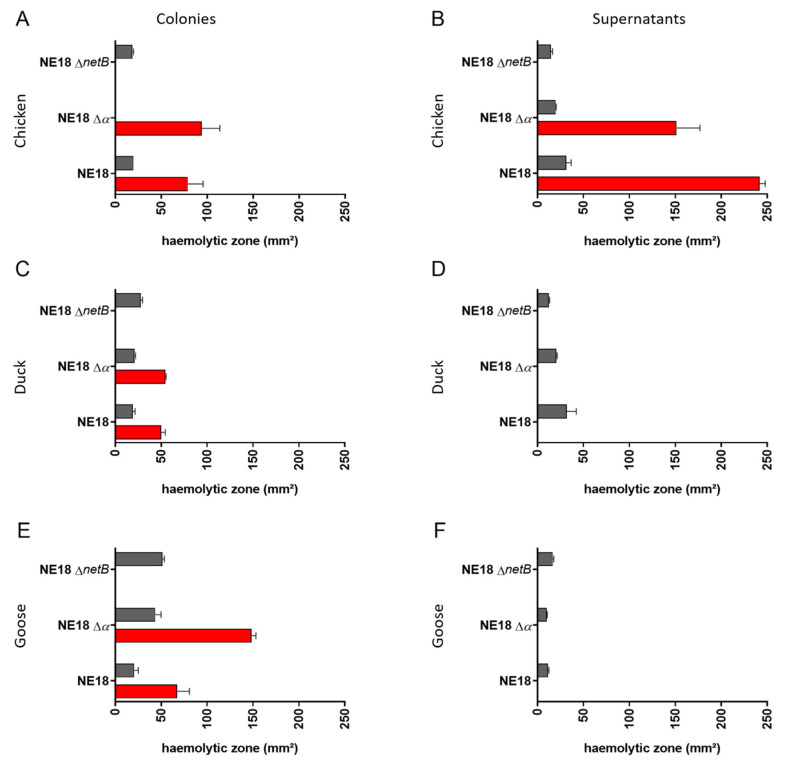
Inner and outer haemolysis induced by *C. perfringens* strain NE18, its NetB mutant (NE18 ∆*netB*) and its alpha toxin mutant (NE18 ∆*α*) on Columbia agar supplemented with blood from different avian species ((**A**,**B**): chicken blood; (**C**,**D**): duck blood; (**E**,**F**): goose blood). Comparison of haemolysis induced from colonies (left column; (**A**,**C**,**E**)) or supernatants (right column; (**B**,**D**,**F**)) was assessed. Grey bars indicate inner haemolysis while red bars represent outer haemolytic zone.

**Figure 3 microorganisms-09-01708-f003:**
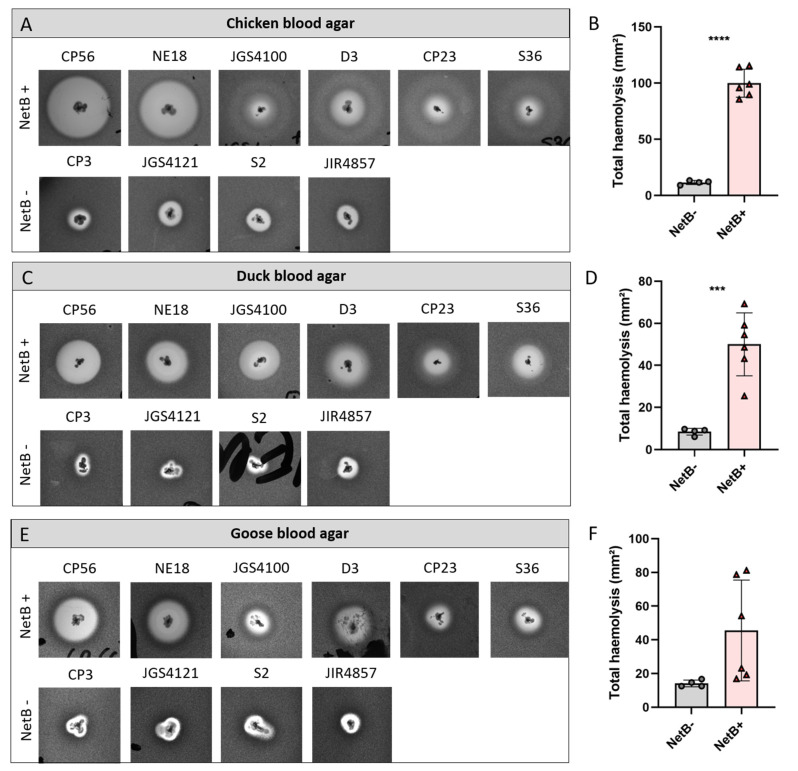
Haemolysis produced by six NetB-positive (CP56, NE18, JGS4100, D3, CP23 and S36) and four NetB-negative *C. perfringens* strains (CP3, JGS4121, S2 and JIR4857) on Columbia agar supplemented with 5% chicken blood (**A**,**B**), duck blood (**C**,**D**) or goose blood (**E**,**F**). The graph shows the total haemolysis induced by each strain (**B**,**D**,**F**). Grey dots represent a NetB-negative strain, red triangles represent a NetB-positive strain. *** *p* < 0.001; **** *p* < 0.0001.

**Figure 4 microorganisms-09-01708-f004:**
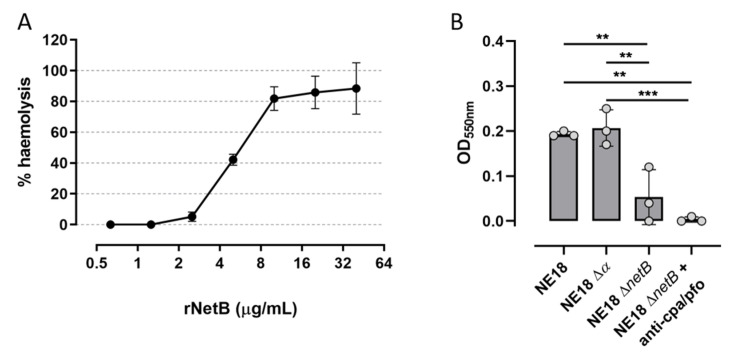
(**A**) Chicken RBC haemolysis induced by a two-fold dilution series of rNetB (starting from 40 µg/mL). The % haemolysis was calculated relative to the negative control (chicken RBCs in HBSS, 0% haemolysis) and positive control (chicken RBCs diluted in dH_2_O, 100% haemolysis) and plotted on a log2 scale against chicken RBCs. (**B**) Comparison of haemolysis induced by the wild-type strain NE18, its alpha toxin mutant (NE18 ∆*α*), the NetB mutant (NE18 ∆*netB*) or NE18 ∆*netB* pre-incubated with anti-alpha toxin and perfringolysin antiserum (anti-cpa/pfo). Haemolysis is observed by an increase in OD due to the release of haemoglobin from the RBCs. ** *p* < 0.01; *** *p* < 0.001.

**Figure 5 microorganisms-09-01708-f005:**
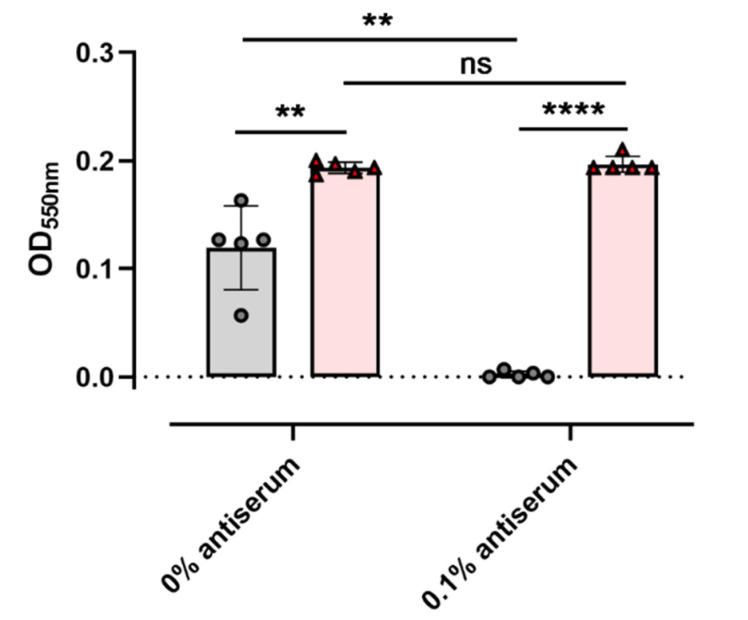
Haemolysis induced by a selection of NetB-positive (red) and NetB-negative (grey) strains, incubated with and without antiserum directed towards *C. perfringens* alpha toxin and perfringolysin O. Haemolysis is observed by an increase in optical density due to the release of haemoglobin from the red blood cells. Individual dots represent individual strains. ** *p* < 0.01, **** *p* < 0.0001, ns = not significant (*p* > 0.05).

**Figure 6 microorganisms-09-01708-f006:**
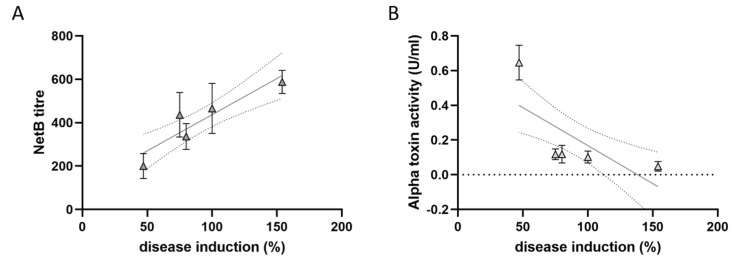
Correlation between the in vitro NetB (**A**) or alpha toxin (**B**) production and the virulence of the *C. perfringens* strains in an NE in vivo model. All strains were grown in TGY medium for 24 h, after which both the NetB titre (supernatant dilution that induces 50% of haemolysis) and alpha toxin activity in the supernatant was determined. The virulence of the strains was previously assessed by Keyburn et al. and is expressed as % disease induction in the NE in vivo model, relative to the virulence of strain NE18 [23]. Each triangle represents the mean ± standard deviation of triplicate experiments. The linear regression line, together with its 95% confidence interval, are plotted in grey.

**Table 1 microorganisms-09-01708-t001:** *C. perfringens* strains used in this study.

Strain ID (Alternative Name)	NetB ^d^	*netB* Sequence ^a^	Description	Origin	Reference
JIR4869 (EHE-NE18)	+	Con ^b^	Necrotic enteritis, broiler	Australia	[25]
JIR12071 (NE18 Δ*α*, NE18-M1)	+	Con ^b^	Alpha toxin mutant from strain JIR4869	Australia	[25]
JIR12331 (NE18 Δ*netB*)	−		NetB toxin mutant from strain JIR4869	Australia	[13]
JIR4860 (EHE-NE5)	+	Con ^b^	Necrotic enteritis, broiler	Australia	[25]
JIR4866 (EHE-NE15)	+	Con ^b^	Necrotic enteritis, broiler	Australia	[25]
JIR12058 (UNK-NE30)	+	G502A (A168T) ^c^	Necrotic enteritis, broiler	Australia	[23]
JIR4857	−		Necrotic enteritis, broiler	Australia	[26]
CP3	−		Healthy broiler	Belgium	[27]
CP4	−		Healthy broiler	Belgium	[27]
CP23	+	G502A (A168T) ^c^	Healthy broiler	Belgium	[14]
CP56 (JIR12037)	+	Con ^b^	Necrotic enteritis, broiler	Belgium	[27]
D3 (99.63206-34)	+	G502A (A168T) ^b^	Necrotic enteritis, broiler	Denmark	[23]
S2	−		Necrotic enteritis, broiler	Denmark	[28]
S36	+	G502A (A168T) ^c^	Necrotic enteritis, broiler	Denmark	-
JGS4100	+	G502A (A168T) ^c^	Necrotic enteritis, broiler	USA	G. Songer, pers.com.
JGS4121	−		Necrotic enteritis, broiler	USA	G. Songer, pers. com.

^a^ *C. perfringens netB* sequence. Con: EHE-NE18 *netB* consensus sequence (EU143239); G502A (A168T): *netB* sequence with a nucleotide change at position 502 of the coding sequence (Change from Ala to Thr at amino acid position 168 of the NetB protein). ^b^ *C. perfringens netB* sequence reported by Keyburn et al. 2010 [23]. The nucleotide substitution reported by Keyburn et al. was numbered relative to the start of the sequencing place (i.e., 267 bp upstream of the *netB* start codon), resulting in G769A. This corresponds to position 502 of the *netB* gene reported by Abildgaard et al. 2010 and this study [24]. ^c^ *C. perfringens netB* sequence obtained in this study. ^d^ Distinction between NetB-positive and NetB-negative *C. perfringens* strains. +: NetB-positive *C. perfringens* strain. −: NetB-negative *C. perfringens* strain.

**Table 2 microorganisms-09-01708-t002:** Overview of the NetB and Alpha toxin production of 5 *C. perfringens* strains with different virulence. All strains were grown in TGY medium for 24 h, after which both the NetB titre (supernatant dilution that induces 50% of haemolysis) and alpha toxin activity (CPA in units/mL) in the supernatants were determined. The virulence of the strains is expressed as % disease induction in an NE in vivo model, relative to the virulence of strain NE18 [23]. Data are expressed as mean ± standard deviation of the mean.

Strain ID	NetB Titre	CPA U/mL	% Disease Induction
JIR12058	588 ± 54	0.048 ± 0.03	154
NE18	466 ± 115	0.102 ± 0.04	100
CP56	337 ± 60	0.119 ± 0.05	80
JIR4866	436 ± 103	0.118 ± 0.03	75
JIR4860	200 ± 57	0.646 ± 0.10	47

## Data Availability

The authors confirm that the data supporting the findings of this study are available within the article and its Appendix A.

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
