# Peer review of "A Rapid and Simple Assay Correlates In Vitro NetB Activity with Clostridium perfringens Pathogenicity in Chickens"

_microorganisms, 2021, doi:10.3390/microorganisms9081708_

Round 1

Reviewer 1 Report

In the manuscript “A rapid and simple assay correlates in vitro NetB activity with Clostridium perfringens pathogenicity in chickens” of Martina Hustá and colelgaues, the authors reported on development of an easy, high-throughput method to determine the NetB activity produced by C. perfringens, which is responsible for necrotic enteritis disease. By using a 96-well plate-based hemolysis assay the NetB activity of C. perfringens culture supernatants could be assessed and correlated with the in vitro NetB activity and virulence of the isolates.

The manuscript provided here provides very interesting data and a very nice possibility for a high-throughput screening. The manuscript is well written, organized and the experiments are well conducted. The manuscript is suitable for publication after only minor revisions. Some examples of style errors are given below. The athors should check the whole manuscript carefully for avoiding such mistakes.

Very nice story and very nice manuscript!!!!

Line 77: 448bp – a space is missing

At several positions 24h – a space is missing

Primer sequences – sometimes written with a space after a triplet, sometimes not – please use a uniform style

Line 95: 4,500

Line 116, 119: please change netb to netB

Line 123: ON - overnight? Please state the abbreviation at the first use

Line 136: Coomassie blue: R or G?

Line 171: 1,000

Author Response

We appreciate the reviewers’ comments and have taken these into account. In this respect, we have revised the manuscript. All changes made to the manuscript are marked with ‘track changes’ and answers to the comments are listed below.

All comments on style errors such as missing spaces, commas and capital letters have been adjusted in the manuscript as suggested by the reviewer and are marked with ‘track changes’.

Comment 1: Primer sequences – sometimes written with a space after a triplet, sometimes not – please use a uniform style
Answer: The primer sequences are now all written without spaces after a triplet.

Comment 2: Line 123: ON - overnight? Please state the abbreviation at the first use
Answer: With the first use in line 87 the abbreviation has been stated.

Comment 3: Line 136: Coomassie blue: R or G?
Answer: Specification of the product had been added in line 138.

Reviewer 2 Report

This paper by Husta et al. presents a simple hemolysis-based in vitro assay for the activity of NetB, the main toxin of Clostridium perfringens (henceforth Cp in this critique), the causative agent of necrotic enteritis in the poultry. The authors have optimized the assay with recombinant NetB toxin, then with NetB-positive and -negative Cp colonies on blood agar plate, and ruled out the roles of two other toxins, namely alpha toxin (CPA) and perfringolysin O (PFO). The research is well-done, logically presented, and convincing. I have a few minor comments and suggestions, as listed below.

(1) The title says the in vitro NetB activity with Cp pathogenicity, and it is also indicated in other places in the paper (Fig. 6, for example), but the concluding sentence in the Discussion (line 468-470) says “No link between the in vitro toxin production and in vivo pathogenicity of the strains was observed, thereby confirming the results from previous studies…”. Please resolve this apparent contradiction and restate as needed.

(2) The authors may want to make a comment on why the pathogenic strains make such diverse amounts of NetB toxin, and discuss the possible reasons, such as regulation at the level of NetB gene expression, high mutation rate in the coding sequence, metabolic or epigenetic regulation etc. 

Author Response

Thank you for reviewing the manuscript entitled “A rapid and simple assay correlates in vitro NetB activity with Clostridium perfringens pathogenicity in chickens”.

We appreciate the reviewers’ comments and have taken these into account. In this respect, we have revised the manuscript. All changes made to the manuscript are marked with ‘track changes’ and answers to the comments are listed below.

Comment 1: The title says the in vitro NetB activity with Cp pathogenicity, and it is also indicated in other places in the paper (Fig. 6, for example), but the concluding sentence in the Discussion (line 468-470) says “No link between the in vitro toxin production and in vivo pathogenicity of the strains was observed, thereby confirming the results from previous studies…”. Please resolve this apparent contradiction and restate as needed.

Answer: The line in the discussion part which the reviewer is referring to, in fact says “No link between the in vitro alpha toxin production and in vivo pathogenicity of the strains was observed, thereby confirming the results from previous studies comparing C. perfringens strains isolated from diseased and healthy poultry flocks.” The focus here is on the alpha toxin not NetB, which otherwise would indeed be a contradiction. Since we are, however, referring to alpha toxin not correlating to the pathogenicity, the conclusion should be clear.

Comment 2: The authors may want to make a comment on why the pathogenic strains make such diverse amounts of NetB toxin, and discuss the possible reasons, such as regulation at the level of NetB gene expression, high mutation rate in the coding sequence, metabolic or epigenetic regulation etc. 

Answer: In the manuscript it was already mentioned that previous studies showed that some AA substitutions in the netB gene influenced the haemolysis against human RBCs (line 435-438). Additional comments on the NetB activity have been added in lines 450-455.